# Difficulties in Emotion Regulation as a Mediator and Gender as a Moderator in the Relationship Between Problematic Digital Gaming and Life Satisfaction Among Adolescents

**DOI:** 10.3390/bs15081092

**Published:** 2025-08-12

**Authors:** İbrahim Erdoğan Yayla, Kübra Dombak, Sena Diril, Betül Düşünceli, Eyüp Çelik, Murat Yildirim

**Affiliations:** 1Department of Guidance and Psychological Counseling, Faculty of Education, Bayburt University, Bayburt 69000, Türkiye; ibrahimeyayla@bayburt.edu.tr; 2Private Sakarya Albatros Special Education and Rehabilitation Center, Sakarya 54100, Türkiye; kubradmbk@gmail.com; 3Ministry of National Education, Bolu 14100, Türkiye; senadiril14@gmail.com; 4Department of Guidance and Psychological Counseling, Faculty of Education, Sakarya University, Sakarya 54100, Türkiye; eyupcelik@sakarya.edu.tr; 5Department of Psychology, Faculty of Science and Letters, Ağrı İbrahim Çeçen Üniversitesi, Ağrı 04100, Türkiye; muratyildirim@agri.edu.tr

**Keywords:** problematic digital gaming, life satisfaction, difficulties in emotion regulation, adolescents, gender differences

## Abstract

**Background:** Problematic digital gaming has emerged as a significant behavioral concern among adolescents, with potential implications for well-being. Understanding the mechanisms through which problematic digital gaming affects life satisfaction and how these mechanisms may differ by gender is important for targeted interventions. **Aims:** This study aimed to investigate the mediating role of difficulties in emotion regulation and the moderating role of gender in the relationship between problematic digital gaming and life satisfaction among adolescents. **Method:** The sample consisted of 458 Turkish adolescents (232 females, 50.7%) aged between 14 and 18 years (M = 16.27). Participants completed the Game Addiction Scale for Adolescents—Short Form, the Life Satisfaction Scale, and the Difficulties in Emotion Regulation Scale—8. **Results:** Problematic digital gaming was found to have a significant negative association with life satisfaction. Difficulties in emotion regulation significantly mediated the relationship between problematic digital gaming and life satisfaction. Furthermore, gender moderated this mediation effect, with difficulties in emotion regulation being more pronounced among males than females. **Conclusions:** The findings highlight the importance of emotional regulation in understanding the negative impact of problematic digital gaming on adolescents’ life satisfaction, particularly among males. These results suggest the need for gender-sensitive approaches in interventions aimed at improving emotion regulation skills and reducing problematic gaming behavior.

## 1. Introduction

Life satisfaction is a subjective indicator of well-being that expresses an individual’s satisfaction with their life. Adolescence, in particular, is a period of rapid and complex changes in terms of physical, cognitive, and emotional development, and is therefore a critical stage in determining life satisfaction ([94]). Research has shown that life satisfaction in adolescence is not merely a temporary state of happiness but also plays a protective role in terms of future psychological health, academic achievement, social relationships, and coping skills ([28]; [67]). Supporting life satisfaction at an early age not only has a protective effect on mental health but also prevents individuals from resorting to negative coping strategies ([20]; [67]). Indeed, studies have shown that adolescents with low life satisfaction are more likely to exhibit aggression, substance use, and risky behaviours ([70]; [73]). Therefore, addressing the concept of life satisfaction during adolescence is important for developing strategies that support mental health at both the individual and societal levels ([34]; [58]; [107]).

## 2. Literature Review

### 2.1. Problematic Digital Gaming and Life Satisfaction

With the rapid advancement of technology in recent years, digital games have become quite widespread, especially among young people, and an indispensable part of daily life. Easy access to mobile devices, strengthening internet infrastructure, and the social interaction opportunities offered by online games are among the main factors increasing the appeal of these games ([100]). Adolescents find digital games appealing; through gaming, they can establish social bonds, escape real-life stress, and meet their self-actualization needs through gaming skills ([71]). However, this appeal also brings with it excessive participation in digital games and problematic usage behaviors ([57]).

Adolescence, in particular, is considered a sensitive developmental stage for problematic digital game playing behaviors because it is a period when individuals struggle with developmental tasks such as identity development, social belonging, and autonomy, experience frequent emotional fluctuations, and are more prone to risky behaviors ([9]; [11]). Problematic digital gaming is defined as an individual’s loss of control over their digital gaming time, disrupting areas such as personal care, work/school, and relationships ([97]). While this condition exhibits an addiction-like pattern, it is not yet officially included as a diagnosis in the American Psychiatric Association’s DSM-5 manual; “Internet Gaming Disorder” is defined only as a condition requiring further study ([2]). Indeed, [49] ([49]) suggests that such behavioral patterns may be related to more general psychological dysfunctions, such as time management, and that early diagnosis may be risky.

Adolescents’ excessive time spent on digital games causes serious behavioral problems ([38]; [123]). Research has found that problematic digital gaming is positively associated with factors such as anxiety ([95]), attention deficit hyperactivity disorder ([60]), obsessive-compulsive disorder ([81]), depression ([85]), and social anxiety ([82]). However, it has been found to be negatively associated with emotion regulation skills ([24]), life satisfaction ([72]; [109]), autonomy, and the need for relationships ([57]). In this context, it can be argued that individuals with problematic digital gaming may experience negative outcomes in many dimensions of their lives, which may also affect their overall life satisfaction levels.

With the proliferation of digital games, problematic gaming has become a problem that causes serious effects on the psychological and social lives of individuals ([124]). This situation is particularly significant among young people and young adults, as it constitutes an important risk factor affecting fundamental well-being, such as life satisfaction ([72]; [92]). Life satisfaction is a cognitive evaluation process that reflects an individual’s satisfaction in different areas of life, such as family, work, friendships, and health ([4]). Adolescence, in particular, is a period of rapid emotional, social, and cognitive development; therefore, life satisfaction is considered an important developmental indicator ([92]). As a matter of fact, studies have emphasized that high life satisfaction in adolescents is closely related to positive variables such as self-esteem ([105]), perception of social support ([6]), school success ([14]), and general psychological resilience ([119]). However, excessive participation in digital games, i.e., problematic gaming, is considered a significant behavioral risk factor that threatens these sources of positive life satisfaction ([54]). Individuals who engage in problematic gaming have been observed to divert time and attention from academic, social, and family responsibilities to the virtual world ([90]). This can lead to weakened ties to real life, social isolation, decreased academic performance, and ultimately, decreased life satisfaction.

A review of the literature shows that uncontrolled digital game playing reduces satisfaction with social relationships and leads to poor psychological well-being ([22]; [59]). Studies conducted in parallel with this finding have also revealed a negative relationship between problematic digital gaming and life satisfaction ([40]; [59]; [72]; [110]). Similarly, [88] ([88]) showed that individuals who play problematic games have lower life satisfaction and higher levels of depression, anxiety, and stress. Research shows that individuals with low life satisfaction turn to digital games more frequently and attempt to address unmet psychological needs through these games ([23]; [43]; [57]). In this context, low life satisfaction serves as a risk factor triggering problematic digital gaming, while the continuation of problematic behavior can further reduce an individual’s life satisfaction, creating a negative cycle ([70]; [72]). Consequently, problematic digital gaming is considered a variable that negatively affects life satisfaction, which is of fundamental importance in the psychosocial development of adolescents.

**Hypothesis** **1.**
*Problematic digital gaming in adolescents negatively predicts life satisfaction.*


### 2.2. The Mediating Role of Difficulties in Emotion Regulation

During adolescence, individuals’ excessive use of digital games can become an escape or coping strategy for dealing with negative emotions. While gaming provides pleasure and distraction in the short term, for some individuals, it can serve as a means of alleviating or suppressing emotional burden in the long term ([43]). This suggests that digital games can become not only entertainment but also an avoidant strategy used to cope with negative emotions ([37]). Attempting to cope with negative emotions in such indirect ways can hinder the development of individuals’ ability to recognize, understand, and regulate their emotions. Turning to external stimuli like games, rather than directly confronting emotions, can undermine the development of healthy emotion regulation skills in the long term ([80]). Indeed, lack of emotional awareness, inadequate expression, and difficulties with regulation strategies have been reported frequently in individuals with problematic digital gaming ([33]; [118]). These findings reveal a strong relationship between problematic digital gaming and difficulties in emotion regulation ([37]).

Difficulty in emotional regulation is defined as the inability to effectively control one’s emotional responses, develop appropriate coping strategies, and cope with negative emotions ([35]; [86]). Recent studies have shown that digital gaming has a detrimental effect on difficulty in emotional regulation. Although problematic digital gaming provide short-term emotional relief, they can weaken emotional intelligence components such as emotional awareness, impulse control, and coping skills in the long term ([74]; [48]). Emotional capacities that are developing during adolescence can be harmed by excessive engagement with digital games ([68]). Furthermore, AI-powered digital games continuously monitor player behavior and use this data to personalize the gaming experience ([30]). These algorithmic feedback loops can impact adolescents’ emotion regulation processes and potentially reinforce addictive behaviors. In support, [111] ([111]) note that AI tools can encourage overreliance on technology, slowing individual responses and impacting individual cognition and performance.

Difficulty in emotional regulation can negatively affect both individual life satisfaction and social relationships. Indeed, studies have shown that difficulty in emotional regulation is positively related to life satisfaction and that difficulty in emotional regulation reduces life satisfaction ([45]; [78]). Research has found that individuals who are unable to regulate their emotions healthily experience more anxiety and depression and, consequently, have lower life satisfaction ([7]; [128]). In this regard, problematic digital gaming indirectly reduces life satisfaction by causing individuals to experience difficulties in regulating their emotions. In summary, while problematic digital gaming is an important factor that negatively affects life satisfaction, difficulty in emotional regulation emerges as an effective mediator variable in understanding this relationship. Additionally, studies indicate that other psychological variables may influence the relationship between problematic digital gaming and life satisfaction. These include variables such as depression ([125]), self-compassion ([55]), and psychological flexibility ([101]). Each of these variables is an important factor influencing life satisfaction, but they are often directly related to emotion regulation skills.

Current research focusing on the relationships between problematic digital gaming, life satisfaction, and difficulties in emotion regulation in adolescents indicates that difficulties in emotion regulation are an important mediating variable in this relationship. [108] ([108]) examined the moderating role of social exclusion and gender in the relationship between family life satisfaction and problematic gaming finding that this relationship is stronger in male adolescents. Similarly, [31] ([31]) examined the relationships between problematic gaming and emotional autonomy and difficulties in emotion regulation using a multiple mediation model, demonstrating that difficulties in emotion regulation are an important mediating variable in predicting problematic gaming. [126]’s ([126]) study emphasized the mediating role of emotion regulation strategies in the relationship between internet addiction and life satisfaction, while [62] ([62]) reported that negative emotions and difficulties in emotion regulation increase internet addiction in Chinese adolescents and that developmental factors are influential in this relationship. [44] ([44]) demonstrated that difficulties in emotion regulation significantly increased social media addiction in women and the severity of problematic gaming in men. [120] ([120]) examined the link between childhood psychological maltreatment and internet problematic gaming among adolescents, and found that maladaptive emotion regulation strategies and depression sequentially mediated this relationship. Studies by [15] ([15]) and [122] ([122]) revealed that emotion regulation strategies have both mediating and moderating effects in the relationships between digital addictions and suicide risk and negative emotions. [127] ([127]) reported that internet addiction predicted insomnia, with difficulties in emotion regulation and anxiety playing a chain mediating role in this relationship, with direct effects being stronger in men and indirect effects being stronger in women. These studies reveal that emotion regulation difficulties are a central psychological variable in the relationship between problematic digital gaming and life satisfaction, and that this relationship is also affected by gender and social factors.

**Hypothesis** **2.**
*Difficulty in emotional regulation mediates the relationship between problematic digital gaming and life satisfaction.*


### 2.3. The Moderating Role of Gender

The relationship between problematic digital gaming and life satisfaction may vary depending on the individual’s psychological characteristics as well as gender differences. Studies show that this relationship is stronger and more negative in males. One of the main reasons for this is that males have lower emotional regulation skills than females ([116]). Emotional regulation skills are critical for an individual’s ability to cope with stress, manage negative emotions, and develop healthy behavior patterns. Girls’ more developed skills in this area may protect them to some extent from the negative effects of problematic gaming on life satisfaction ([115]). A study found that female gamers regulate their emotional responses more harmoniously and thus feel the effects of problematic gaming on life satisfaction to a lesser extent ([116]). In addition, males spend more time on games and exhibit higher levels of problematic behaviour ([93]). One study found that male adolescents are at a significantly higher risk of problematic digital gaming than females, negatively impacting academic achievement, sleep quality, and social relationships ([46]). Similarly, [26] ([26]) highlighted a strong correlation between problematic digital gaming and increased impulsivity and lower self-regulation skills in male adolescents. This situation may lead to greater harm to their life satisfaction ([41]; [125]). Furthermore, when men play games primarily for competition, success, and escape motivations, combined with deficiencies in emotional regulation skills, this can elevate addiction to a more dysfunctional level ([37]; [116]). All these findings indicate that gender plays a moderating role in the relationship between problematic gaming and life satisfaction, particularly in male individuals, where this relationship is stronger and more negative ([115]). Girls’ advanced emotion regulation skills make them more resilient to the psychological consequences of digital game use, while boys’ weakness in this skill amplifies the negative effects of problematic gaming on life satisfaction ([36]; [80]).

**Hypothesis** **3.**
*Difficulty in emotional regulation fully mediates in boys, while partially mediating in girls.*


### 2.4. Present Study

Today, problematic digital gaming has significant consequences for psychological well-being, especially among young adults. Although problematic gaming behavior has been examined through various psychological variables in the context of adolescents, research on the adolescent population remains limited ([84]; [99]). Problematic digital gaming is observed with increasing frequency, particularly during adolescence and young adulthood, and multidimensionally impacts individuals’ psychological well-being. In this context, its relationship with positive psychological outcomes, such as life satisfaction, gains greater importance from the perspective of psychological resilience and well-being ([115]). However, the psychological factors underlying the relationship between problematic gaming and life satisfaction remain poorly elucidated. This study proposes a unique model to address this gap by considering difficulty in emotion regulation as a mediating variable. Theoretically, this research makes an integrated contribution to the problematic gaming literature using a positive psychology approach. Existing studies generally focus on the relationships between problematic gaming and psychopathological symptoms (depression, anxiety, anger) ([42]; [121]), while subjective well-being indicators, such as life satisfaction, are neglected. Furthermore, while emotion regulation difficulties are generally considered an independent risk factor ([99]), this study models them as a structural mediating mechanism. In this respect, the study offers a new theoretical perspective by examining the indirect paths between problematic gaming and life satisfaction.

Methodologically, the study goes beyond previous correlational or regression analyses ([37]; [113]) by employing structural equation modeling. This approach offers a more advanced framework than previous methodological approaches in the literature, explaining the complex relationships between variables and testing the impact of the mediating variable more robustly. In terms of sampling, this study is one of the few studies specifically in the Turkish context that examines the relationship between problematic gaming and life satisfaction. In this respect, the study fills a significant representation gap in the literature by being conducted with a culturally underrepresented population. In the existing literature, the role of gender in these relationships has largely been treated as a secondary control variable ([113]; [63]); the psychological impacts of digital gaming experiences among male and female adolescents have often not been analyzed in detail ([10]; [11]). This study fills this important gap in the literature by incorporating gender into the model not merely as a demographic variable but as a determinant factor affecting emotion regulation processes and life satisfaction. Male adolescents’ lower levels of emotion regulation skills and higher risk of addiction make the moderating role of gender critical in understanding the effects of addiction on life satisfaction ([116]). In this context, the study conceptually distinguishes itself from similar previous models by incorporating gender into emotion regulation-based explanations and offers a unique contribution. Therefore, the study’s findings may contribute to the development of psychoeducational programs that strengthen emotional awareness and regulation skills to prevent problematic gaming. Furthermore, they may offer parents, educators, and mental health professionals the potential to offer more effective and holistic intervention strategies regarding adolescents’ digital game playing. The model visualizing the current study hypotheses is presented in Figure 1.

## 3. Method

### 3.1. Participants and Procedure

The study included 458 Turkish adolescents. A total of 226 (49.3%) of the participants were male, while 232 (50.7%) were female. The ages of the participants ranged from 14 to 18, with a mean age of 16.27 years. A total of 136 (29.7%) of the participants were in the 9th grade, 139 (30.3%) were in the 10th grade, 104 (22.7%) were in the 11th grade, and 79 (17.2%) were in the 12th grade. After obtaining permission from the University Ethics Committee, schools were selected based on their accessibility within the city center and their willingness to collaborate in the data collection process. Specifically, among ten public high schools contacted, two were chosen due to their convenient location for the research team and the cooperation of school administrators who granted permission for the study to be conducted on-site. Within this framework, selected high schools were visited in April 2025. After students were informed about the study, those who volunteered were told that they could participate only after obtaining written parental consent, as they were under the age of 18. In addition, assent was also obtained from each adolescent participant, ensuring that their participation was based on their own voluntary agreement. They were also informed that there was no incentive and that the data were confidential and anonymous. Three days after these briefings, schools were revisited to collect responses. Data were not collected from students who did not complete 80 percent of the questionnaires, did not agree to participate, or whose parents did not sign the consent form. Students’ responses were not linked to any personal identifiers. Of the approximately 550 students who received the invitation, 458 completed forms were accepted, resulting in a response rate of 83.3%. No reminders or follow-up incentives were used to increase the response rate. Missing data were analyzed using frequency and pattern analysis in SPSS, and univariate outliers were assessed using standardized z-scores (|z| > 3.29). No missing values or univariate outliers were found. In conclusion, all participant data were within acceptable ranges, and no data were excluded from the analysis.

### 3.2. Instruments

**The Game Addiction Scale for Adolescents-Short Form:** It was developed by [3] ([3]) to determine the problematic gaming levels of adolescents in terms of their gaming behaviour. The scale consists of a total of 9 items, a single dimension, and a Likert-type rating ranging from 1 to 5. As the scores obtained from the scale increase, the level of problematic gaming increases. The reliability value of the scale is α = 0.81. Additionally, the scale’s fit indices are χ^2^ (27) = 2.514, CFI = 0.96, GFI = 0.97, RMSEA = 0.054, and NFI = 0.94 ([31]).

**Life Satisfaction Scale:** It was used to determine the level of satisfaction of adolescents with their general life experiences. The scale (Original: [29] ([29]), Turkish: [25] ([25])) consists of five items, a single dimension, and a 5-point Likert scale. It has been stated that as the scores obtained from the scale increase, the level of satisfaction of adolescents also increases. The confirmatory factor analysis (CFA) of the adaptation to Turkish culture showed good fit (χ^2^/df = 1.17, RMSEA = 0.030, AGFI = 0.97, CFI = 1.00, SRMR = 0.019, GFI = 0.97). The Cronbach’s Alpha value of the scale is 0.88 ([56]).

**Difficulties in Emotion Regulation Scale-8:** It has been used to determine the emotional regulation difficulties experienced by adolescents. The Scale Scale (Original: [89] ([89]), Turkish: [32] ([32])) has 8 items, 4 factors (purpose, impulse, rejection, and strategy), and a 5-point Likert scale. Higher scores indicate greater difficulty. Cronbach’s alpha coefficients range from 0.87 for the entire scale to 0.68 and to 0.77 for the subscales. The results of the DFA analysis indicate that the general fit indices (χ^2^/df = 3.05; GFI = 0.97; NFI = 0.96; TLI = 0.95; CFI = 0.98; RMSEA = 0.075) are adequate ([32]).

### 3.3. Data Analysis

SPSS 27.0 and AMOS 24 statistical software packages were used for the analysis of the research. First, the normality of the data was determined by examining the normal distribution graph, skewness, and kurtosis values. Then, Pearson Correlation analysis was conducted to determine the relationships. Structural equation modeling (SEM) was conducted using AMOS 24 to examine the hypothesized relationships between variables and to test for mediation and moderating mediation effects. A two-stage modeling approach was applied: first, the relationship between the independent variable and the dependent variable was established, and then a structural equation model was constructed by including the mediating variable in the model. Model fit indices (e.g., CFI, TLI, RMSEA, SRMR) were used to assess the adequacy of the models.

A multigroup SEM approach was used to test whether the indirect effect of problematic digital gaming on life satisfaction through difficulties in emotion regulation varies by gender. Prior to multigroup comparisons, measurement invariance across genders was assessed. Both structural and metric invariance were found, indicating that factor structure and factor loadings were equivalent between the male and female groups. Critical ratio comparisons were then used to assess whether path coefficients differed significantly between groups.

## 4. Results

### 4.1. Preliminary Analyses

Table 1 shows that life satisfaction is negatively correlated with problematic digital gaming (*r* = −0.45, *p* < 0.001), goal (*r* = −0.54, *p* < 0.001), impulse (*r* = −0.53, *p* < 0.001), non-acceptance (*r* = −0.50, *p* < 0.001), and strategy (*r* = −0.58, *p* < 0.001). Problematic digital gaming is positively related to goal (*r* = 0.41, *p* < 0.001), impulse (*r* = 0.45, *p* < 0.001), non-acceptance (*r* = 0.48, *p* < 0.001), and strategy (*r* = 0.39, *p* < 0.001). Impulse, one of the subscales of difficulties in emotion regulation (*r* = 0.70, *p* < 0.001), is positively related to acceptance (*r* = 0.70, *p* < 0.001) and strategy (*r* = 0.78, *p* < 0.001). One of the subdimensions of difficulties in emotion regulation, impulse, is positively related to acceptance (*r* = 0.71, *p* < 0.001) and strategy (*r* = 0.72, *p* < 0.001). Acceptance is positively related to strategy (*r* = 0.72, *p* < 0.001).

The results presented in Table 1 show that the skewness values range from 0.64 to −0.17, while the kurtosis values range from −1.18 to −0.43. These values are between +1.5 and −1.5, which is within the acceptable range for normality ([106]). Before testing the mediating model, the level of problematic digital gaming’s impact on life satisfaction was determined. The findings are presented in Figure 2.

Figure 2 shows the association between problematic digital gaming and life satisfaction was (*β* = −0.51, *p* < 0.001). The fit indices for Figure 2 were found to be adequate with χ^2^/df = 4.60, CFI = 0.932, GFI = 0.897, TLI = 0.932, and SRMR = 0.049 ([8]; [52]; [53]; [75]; [117]). Prior to testing the structural model across gender, measurement invariance was examined using a multi-group CFA approach. The configural invariance model showed acceptable fit across gender groups (χ^2^(264) = 693.68, CFI = 0.93, RMSEA = 0.060, TLI = 0.91). Metric invariance was also supported, as the model constraining factor loadings to be equal did not significantly worsen model fit compared to the configural model (Δχ^2^(15) = 16.78, *p* = 0.332; ΔCFI = 0.003). These results indicate that structural paths can be meaningfully compared across gender groups.

### 4.2. Testing for Moderated Mediation

The structural equation model created by including difficulties in emotion regulation as a mediator in the relationship between problematic digital gaming and life satisfaction is presented in Figure 3 for boys and Figure 4 for girls. The fit indices for both models are shown in Table 2.

When the findings presented in Figure 2 (Stage One) of the model showing the mediating effects of the study are compared with those presented in Figure 3 (Stage Two), the relationship between problematic digital gaming and life satisfaction, when difficulty in emotional regulation is included as a mediating variable, was found to be *β* = −0.12 for boys. The relationship between problematic digital gaming and life satisfaction decreased from *β* = −0.51 to *β* = −0.12 in boys and became insignificant (*p* > 0.05).

When the findings presented in Figure 2 (Stage One) of the model showing the mediating effects of the study are compared with those presented in Figure 4 (Stage Two), the relationship between problematic digital gaming and life satisfaction, when difficulty in emotional regulation is included as a mediating variable, was found to be *β* = −0.33 for girls. The relationship between problematic digital gaming and life satisfaction decreased from *β* = −0.51 to *β* = −0.33 in girls, and its predictive power weakened (*p* < 0.001). Thus, the effect of problematic digital gaming on life satisfaction became statistically insignificant in males (full mediation) when difficulty in emotional regulation was included in the model, while the relationship weakened (partial mediation) but did not disappear completely in females. Additionally, when comparing gender differences (Critical Ratios for Differences), the differences between male and female groups were significant (*β*_girls_ = −0.33, *β*_boys_ = −0.12, C.R. = 2.560 > 1.96, *p* < 0.001). Acceptance ranges for fit indices specified by relevant statistics experts in studies ([8]; [52]; [53]; [75]; [117]) and the results of the fit indices obtained for the mediator model tested in the current study are demonstrated in Table 2.

Examining the fit indices for the mediator model as seen in Table 2, the fit indices of the current research model are at an acceptable level. Table 3 presents the direct, indirect, and total effects of problematic digital gaming on life satisfaction in both boys and girls, where the mediation effects were evaluated.

Figure 5 shows that gender moderates the relationship between problematic digital gaming and life satisfaction. The negative association is stronger in females, suggesting that increased problematic digital gaming leads to a greater decrease in life satisfaction among girls compared to boys.

## 5. Discussion

This study examined the relationship between psychological mechanisms affecting adolescents’ life satisfaction, namely problematic digital gaming and difficulty in emotional regulation, and investigated the moderating role of gender. The findings revealed that difficulty in emotional regulation mediates the relationship between problematic digital gaming and life satisfaction in adolescents. However, this mediating effect was found to be fully mediating in males, while only partially mediating in females. The study results were evaluated in light of the relevant literature.

### 5.1. The Relationship Between Problematic Digital Gaming and Life Satisfaction

This study confirms that problematic digital gaming is one of the psychological mechanisms that negatively predict adolescents’ life satisfaction levels, as in many previous studies. In other words, as the level of problematic digital gaming increases among adolescents, the level of life satisfaction decreases ([50]; [72]). When the background of the relationship between problematic digital gaming and life satisfaction is examined, it has been observed that excessive gaming is associated with a decrease in social skills and an increase in loneliness levels in adolescents over time, and this is proportional to low life satisfaction ([69]). The negative correlation of problematic digital gaming with physical health indicators such as sleep quality and eating habits is another factor associated with low life satisfaction ([18]; [61]; [129]). It is known that young people with problematic digital gaming behaviour spend most of their time playing digital games ([98]). However, the strong correlation between time spent playing games and low study motivation and academic achievement is associated with low life satisfaction ([112]). In addition, the relationship between excessive preoccupation with games and family problems has also been found to be associated with a decrease in the life satisfaction of adolescents ([27]). Another factor associated with low life satisfaction in adolescents is the strong link between problematic digital gaming in adolescents and intense mental health problems such as depression, stress, and anxiety ([85]). In summary, problematic digital gaming has been found to be closely associated with low life satisfaction in adolescents, with negative correlations with social relationships, physical and mental health, and academic achievement. Furthermore, studies also point to the relationship between low life satisfaction and high problematic digital gaming ([40]; [87]). This suggests that adolescents with low life satisfaction may turn to gaming as a coping mechanism and may enter a vicious cycle over time.

### 5.2. The Mediating Role of Difficulties in Emotion Regulation Between Problematic Digital Gaming and Life Satisfaction

When the relationship between problematic digital gaming and difficulties in emotion regulation, which is the first pathway of the mediation model, was examined, it was found that as the level of problematic digital gaming increased, the level of emotion regulation difficulties increased in adolescents. This finding is consistent with the relevant literature ([37]; [51]; [64]). Adolescents may turn to digital games to avoid facing negative emotions such as anger, anxiety, stress, and loneliness ([96]). After a while, excessive gaming makes it difficult to cope with emotions in a healthy way and weakens emotion regulation skills ([99]). Excessive engagement with digital games is closely associated with sleep problems, academic difficulties, and poor time management in daily life ([5]; [79]). This may be accompanied by emotional instability and difficulty in emotional regulation.

The second path of the mediation model in the current study revealed that emotion regulation difficulties negatively predicted life satisfaction. In other words, as the level of emotion regulation difficulties increases in adolescents, their life satisfaction levels decrease. Adolescence is a complex period involving many developmental tasks, and emotional fluctuations and changes are particularly common during this period ([1]; [77]). During this critical period, adolescents with poor emotion regulation skills may feel overwhelmed during emotional fluctuations and difficulties in coping with developmental tasks, which may be associated with low life satisfaction ([39]; [83]). Furthermore, adolescents who have difficulty coping with emotions in a healthy way are reported to be prone to substance abuse such as smoking and alcohol ([92]). Adolescents who struggle with emotion regulation may be more prone to loneliness due to difficulties with social skills such as forming close friendships and expressing themselves in social settings ([66]). A similar impact seen in negative social relationships is also observed in the academic sphere. Adolescents with poor emotion regulation skills report a lack of ability to maintain and develop academic motivation. Lack of concentration is also associated with poor academic performance ([114]). Furthermore, adolescents who struggle with emotion regulation have been reported to be prone to many psychological problems such as anxiety and depression ([19]; [130]). Consequently, difficulties in emotion regulation in adolescents may be closely related to loneliness, psychological problems, academic failure, risk of substance abuse, and poor impulse control, as well as low life satisfaction, which can be predictive of poor motivation. When these findings are considered alongside the mediating effect in the current study, problematic digital gaming appears to directly predict life satisfaction and is also indirectly related to life satisfaction through predicting difficulties in emotion regulation. Another finding from the mediation analysis indicates that difficulties in emotion regulation have a greater predictive power on life satisfaction during adolescence than problematic digital gaming. Based on the preceding findings, difficulties in emotion regulation seem to exert a more generalized influence on adolescents’ well-being compared to problematic digital gaming, especially in relation to life satisfaction.

### 5.3. The Moderating Role of Gender

In the current study, the moderating effect of gender on the indirect relationship between problematic digital gaming and life satisfaction through difficulties in emotion regulation was examined. The results indicated that difficulties in emotion regulation play a full mediating role in male adolescents, while a partial mediating effect was observed in female adolescents. In other words, the effect of problematic digital gaming on life satisfaction in males is superseded by difficulties in emotion regulation, whereas the direct effect persists, albeit weakly, in females. Drawing on the literature on gender socialization and emotional development, girls are generally found to exhibit higher emotional intelligence and awareness ([47]; [131]). Therefore, the impact of problematic gaming on difficulties in emotion regulation may be weaker in girls than in boys. While boys exhibit more eager and passionate player profiles in digital games, girls exhibit more avoidant and reluctant player profiles ([12]). Based on this, it can be expected that boys’ levels of difficulties in emotion regulation in the effect of problematic digital gaming will be higher than girls’. The change in the mediating effect can also be explained by the gender differences in emotion expression patterns. While girls are known to express their emotions more freely and talk more about their feelings, boys are reported to suppress their emotions and adopt a more introverted attitude toward expressing emotions ([16]; [21]). In addition to this information, neurobiological research reveals that women are more active in higher-order regulatory brain regions such as the prefrontal cortex, whereas men respond more strongly in structures associated with emotional responses, such as the amygdalan ([17]). These differences may partly explain women’s tendency to use more effective regulation strategies in emotionally stimulating situations ([76]; [91]). In summary, it can be said that difficulties in emotion regulation have a more central effect on explaining the relationship between problematic digital gaming and life satisfaction in males than in females.

### 5.4. Contribution and Limitations

The study results have introduced new perspectives on intervention and prevention activities aimed at reducing the risk of low life satisfaction in adolescence. First, it has revealed that problematic digital gaming negatively predicts life satisfaction in adolescents. Therefore, to reduce the risk of low life satisfaction, school psychological counseling services that are easily accessible to adolescents should implement awareness and intervention strategies focusing on problematic digital gaming behaviors—particularly those associated with functional impairments and emotional difficulties—without framing such behaviors as a clinically validated psychiatric diagnosis. Second, it was observed that difficulties in emotion regulation negatively predict life satisfaction in adolescents. As an effective effort to reduce the risk of low life satisfaction in adolescents, priority should be given to awareness activities that develop emotion regulation skills. The third and perhaps most important contribution of this study is that it emphasizes the need to consider gender differences in all prevention and intervention activities aimed at reducing the risk of low life satisfaction. In particular, it is important not to overlook the need to support male adolescents in recognizing, expressing, and becoming more aware of their emotions.

As with any study, this study has certain limitations. First, since the data were collected at a single point in time, i.e., this is not a longitudinal study, it is not possible to establish a complete causal relationship between the variables in the current study. Future studies would benefit from testing the existing relationships using longitudinal designs and experimental models. In addition to the current findings, recent research demonstrates the value of more advanced analytic approaches: latent profile analysis (e.g., [104]), person-centered modeling via mixture clustering ([102]), and mixed-methods designs integrating qualitative and quantitative insights ([103]) to better capture individual differences in adolescents’ gaming and emotional experiences. Second, the study data were collected from adolescents through self-reporting. This single-source data collection approach may increase the risk of methodological bias, particularly for subjective and situational variables such as emotion regulation, and may not reflect all aspects of participants’ behavior ([66]). Since this situation carries the risk of respondent bias ([65]), it is recommended that future studies collect data from multiple sources, such as parents, teachers, or peers, to increase the validity of the findings. Third, this study sample is limited to students from different types of high schools in a specific city in Turkey. Although this study provides important findings specific to Turkish adolescents, caution should be exercised when generalizing considering differences in cultural norms, parental attitudes, and digital access. Replication in different cultural contexts is recommended to test the validity of the findings. Fourth, the study assumes problematic digital gaming as a distinct diagnostic entity. However, it should be noted that there is currently no global consensus on the clinical validity of problematic digital gaming ([13]). Therefore, the interpretation of the results should be considered within the context of this ongoing debate.

### 5.5. Conclusions

This study examined the mediating effect of emotion regulation difficulties and the moderating role of gender in the relationship between problematic digital gaming and life satisfaction among adolescents. The findings revealed that emotion regulation difficulties fully mediated the relationship between problematic digital gaming and life satisfaction among males. On the other hand, emotion regulation difficulties in females partially mediated the relationship between problematic digital gaming and life satisfaction. These results reveal that emotion regulation difficulties are a decisive psychological mechanism in the effect of problematic digital gaming on life satisfaction in adolescents and that this relationship varies according to gender. The findings indicate the importance of addressing emotion regulation difficulties in interventions targeting life satisfaction, especially in boys, while suggesting that other psychosocial factors should also be included in the process for girls. Future studies are recommended to include different psychosocial variables (social support, empathy, etc.) in addition to the model and to plan longitudinal designs. Furthermore, gender sensitivity should not be overlooked when developing preventive and developmental strategies for mental health services aimed at life satisfaction.

## Figures and Tables

**Figure 1 behavsci-15-01092-f001:**
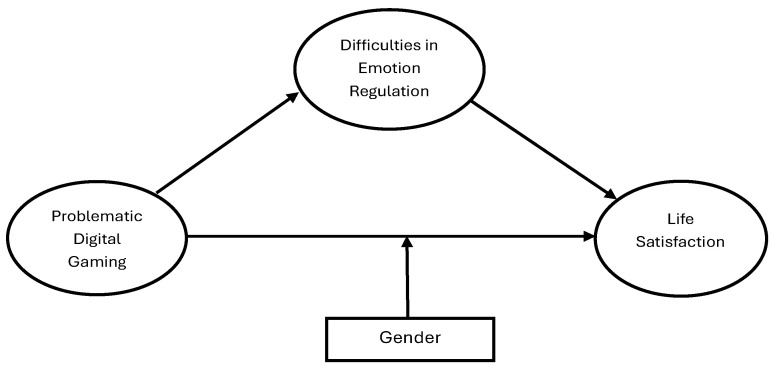
The proposed hypothetical model developed by the authors.

**Figure 2 behavsci-15-01092-f002:**
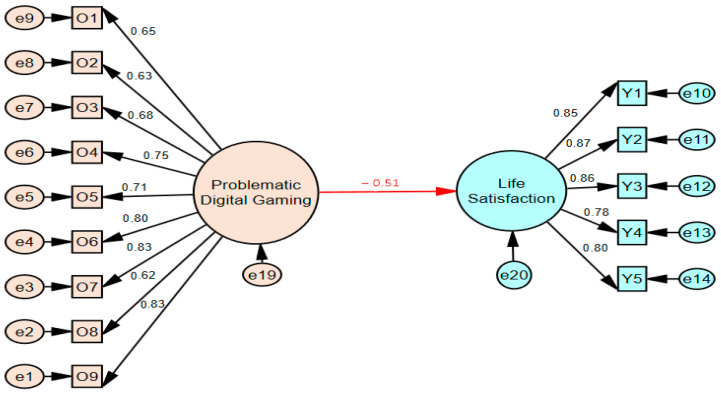
The relationship between problematic digital gaming and life satisfaction (Stage 1).

**Figure 3 behavsci-15-01092-f003:**
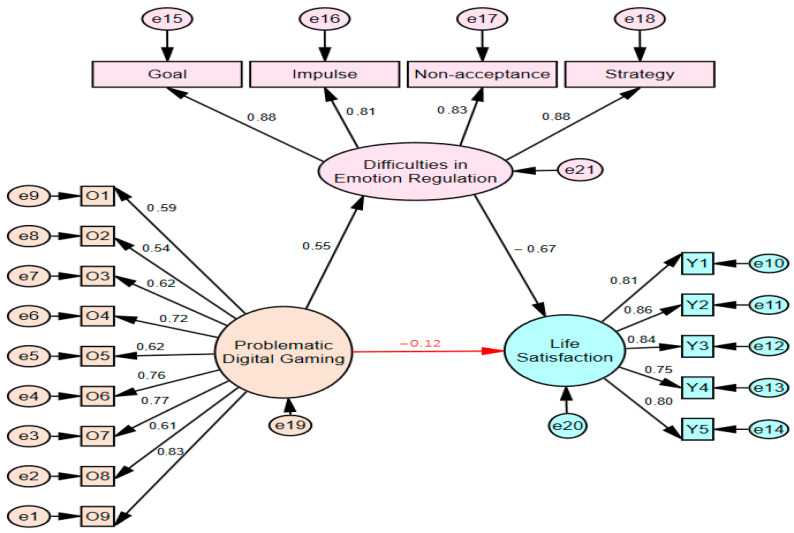
Moderated mediation effect of boys (Stage 2).

**Figure 4 behavsci-15-01092-f004:**
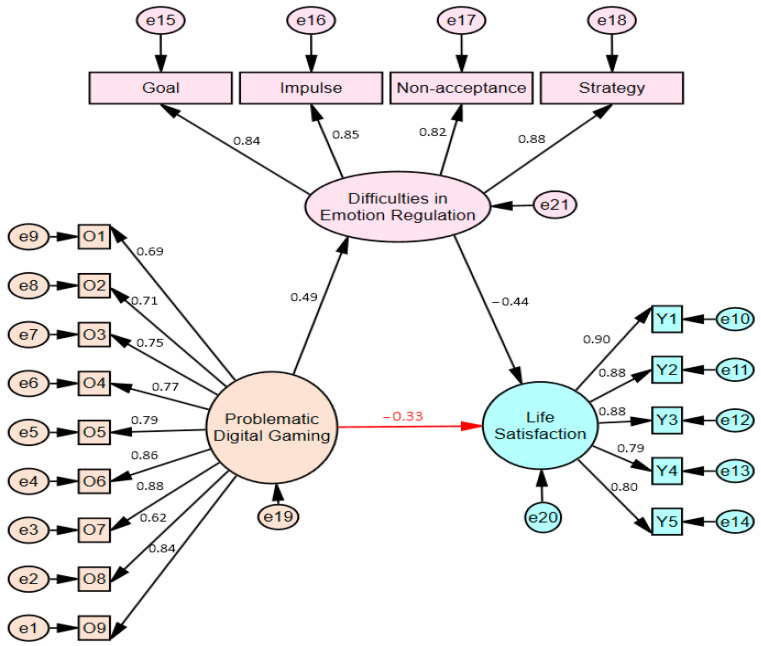
Moderated mediation effect of girls (Stage 2).

**Figure 5 behavsci-15-01092-f005:**
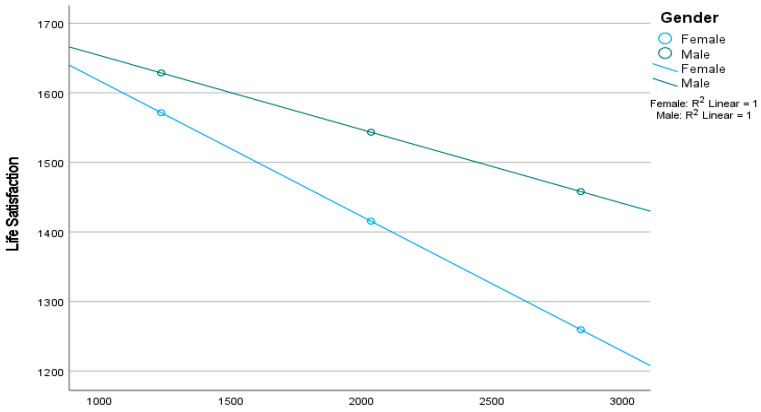
Gender moderated the relation between problematic digital gaming and life satisfaction.

**Table 1 behavsci-15-01092-t001:** The results of correlation analysis and descriptive statistics.

**Variable**	**1**	**2**	**3**	**4**	**5**	**6**
1. Life Satisfaction	1					
2. Problematic Digital Gaming	−0.455 **	1				
3. Goal	−0.538 **	0.407 **	1			
4. Impulse	−0.526 **	0.454 **	0.700 **	1		
5. Non-acceptance	−0.499 **	0.480 **	0.698 **	0.707 **	1	
6. Strategy	−0.567 **	0.387 **	0.779 **	0.717 **	0.723 **	1
Mean	14.83	20.42	5.85	5.52	4.98	5.44
SD	5.49	8.02	2.10	2.36	2.35	2.29
Skewness	−0.17	0.64	0.30	0.37	0.48	0.36
Kurtosis	−1.18	−0.43	−0.81	−1.00	−0.92	−0.85

** *p* < 0.01.

**Table 2 behavsci-15-01092-t002:** Acceptance ranges for fit indices and fit indices obtained from the mediator model test in females and males.

İndices	Perfect Fit Limit	Acceptable Fit Limit	Scale Indices	Result
X^2^/DF	0–2.5	≤5	2.62	Acceptable
RMSEA	≤05	≤08	0.06	Acceptable
SRMR	≤05	≤08	0.05	Perfect
CFI	≥95	≥90	0.92	Acceptable
GFI	≥90	≥85	0.85	Acceptable
IFI	≥95	≥90	0.92	Acceptable

**Table 3 behavsci-15-01092-t003:** Direct and indirect effects of problematic digital gaming on life satisfaction.

Gender	Effect Type	Path	*β* and *p*-Value
Female	Direct Effect	Problematic Digital Gaming → Life Satisfaction	−0.33, <0.001
	Indirect Effect	Problematic Digital Gaming → Difficulties in Emotion Regulation → Life Satisfaction	−0.18, <0.001
	Total Effect	Combined	−0.51, <0.001
Male	Direct Effect	Problematic Digital Gaming → Life Satisfaction	−0.12, >0.001
	Indirect Effect	Problematic Digital Gaming → Difficulties in Emotion Regulation → Life Satisfaction	−0.39, <0.001
	Total Effect	Combined	−0.51, <0.001

## Data Availability

The data that support the findings of this study are available from the corresponding author upon reasonable request. The data are not publicly available due to the inclusion of raw materials used exclusively for the analyses conducted in this research.

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
