# Peer review of "Difficulties in Emotion Regulation as a Mediator and Gender as a Moderator in the Relationship Between Problematic Digital Gaming and Life Satisfaction Among Adolescents"

_behavsci, 2025, doi:10.3390/bs15081092_

Round 1

Reviewer 1 Report

Comments and Suggestions for Authors

A study aiming to inform and advise school psychologists of the corroborated results of three hypotheses: H1: Digital game addiction in adolescents negatively predicts life satisfaction.

H2:  Difficulty in emotional regulation mediates the relationship between digital game addiction and life satisfaction.

H3: Difficulty in emotional regulation fully mediates in boys, while partially mediating in girls.

The work is well-written, well-conceptualized, and well-analyzed, and the conclusions follow from the results. The work is replicable because of the sufficient methodological details.

There two weakness. Unfortunately, the first weakness affects the intended aim of advising school psychologists. The weakness is that the Literature review significantly relies on outdated research and begins with the assumption that digital game addiction is an addiction without examining the contrary evidence. The author must redo this Literature Review to focus on research published since 2021 and consider both sides of whether there is digital game addiction in adolescents, especially in boys.

With this change, the authors must rework their recommendation in lines 383–385 regarding what they should advise school psychologists. The advice must be a balance regarding the current debate regarding the diagnosis of digital game addiction. An additional limitation is that the study assumes digital game addiction to be an accepted diagnosis. Please include that there is no consensus on the existence of digital game addiction.

The second weakness is that, in not considering the most recent research on the topic sufficiently, the authors have not demonstrated the uniqueness of their study. Here is a Google Scholar search of the topic for research published since 2021: https://scholar.google.ca/scholar?hl=en&as_sdt=0%2C5&q=Difficulties+in+Emotion+Regulation+as+a+Mediator+and+Gender+as+2+a+Moderator+in+the+Relationship+Between+Digital+Game+3+Addiction+and+Life+Satisfaction+Among+Adolescents&btnG=. Note that there are “About 17,500 results”. The authors must read the most relevant of these and include a section on what their study adds to the literature.

Author Response

Please see the attached file. All our responses to the reviewer 1 comments have been included in the PDF document.

Reviewer 2 Report

Comments and Suggestions for Authors

This manuscript investigated an important issue among adolescents. However, I have some concern on the analysis and results.

  1. For the instruments, it is good to see that the authors presented some psychometric properties. Please clarify if these values are from the current study or from the literature.
  2. Please analyze and report the direct and indirect effects.
  3. For the statement “Figure 2 shows the association between digital game addiction and life satisfaction was (r = -.51, p < .001).”, it appears that you are reporting the results from SEM, and it should be β instead of r. Please clarify.
  4. Some statements for reporting the results are inappropriate. For example, “The relationship between digital game addiction and life satisfaction decreased from r = -.51 to β = -.12”. Please revise.

Author Response

Please see the attached file. All our responses to the reviewer 2 comments have been included in the PDF document.

Reviewer 3 Report

Comments and Suggestions for Authors

The manuscript explores the mediating role of emotion regulation and the moderating role of gender in the association between digital game addiction and life satisfaction among adolescents. This is a highly relevant topic in the current psychological landscape, particularly given the pervasive role of digital environments in adolescent development. The authors should be commended for addressing a question with practical and theoretical implications. However, several critical aspects of the manuscript require substantial revision to enhance its rigor, clarity, and contribution to the field.

  1. Novelty and Contribution to the Literature

While the topic remains significant, the manuscript does not clearly articulate what its added value is in comparison to the existing literature. The mechanisms linking gaming addiction to life satisfaction, including the roles of emotional dysregulation and gender, have been investigated in prior studies using similar models. To strengthen the manuscript's contribution, the authors should clearly justify the need for this study, specifying whether the model they propose introduces new theoretical perspectives, improves on prior methodological approaches, or fills a gap in under-researched populations. At present, the study reads more as a replication or local validation than as a conceptual advance. Including recent conceptual work on how digital environments alter emotional development, such as through AI-mediated interactions or algorithmic feedback loops, could add depth (see Di Plinio, 2025, Social Sciences & Humanities Open, https://www.sciencedirect.com/science/article/pii/S2590291125001615).

  1. Theoretical Framing and Hypotheses

The literature review is comprehensive but overly descriptive and lacks integrative synthesis. It proceeds as an accumulation of evidence rather than as a theoretically driven rationale that guides the reader toward the hypotheses. This issue is especially evident in the transition between the literature review and the formulation of hypotheses, which are stated in very concise and isolated form, without sufficient embedding in the theoretical framework. Each hypothesis should be more thoroughly contextualized. For instance, if the authors predict a full mediation effect for males and a partial mediation for females, they should explain why this pattern is expected based on developmental psychology or emotion regulation theory. Similarly, the rationale behind selecting emotion regulation (rather than, say, self-control or psychological resilience) as the mediator is not fully developed. A clearer articulation of the model’s conceptual underpinnings is needed.

  1. Methodology and Data Collection

While the description of the sample and instruments is acceptable, the section on participants and procedure is too brief and lacks essential details. It remains unclear how the schools were selected, what recruitment process was followed, how consent was obtained in practice, and whether any exclusion criteria were applied. Additionally, no information is provided regarding missing data, handling of outliers, or control for common method bias. These elements are especially relevant in adolescent research, where consent processes, potential response biases, and school-level effects may significantly influence results. The authors should provide a clearer narrative of how the data were collected logistically, including whether responses were anonymous, if incentives were used, and how response rates were managed.

  1. Statistical Analyses and Reporting

4a. The statistical analysis plan is underdeveloped. The Data Analysis section merely outlines the software used and the general modeling approach. However, the Results section introduces structural equation modeling, moderated mediation analysis, and critical ratio comparisons—none of which are properly introduced or justified in the methods. This disconnect undermines the transparency of the analytic process. If moderated mediation was tested using multi-group SEM, the authors should specify whether they assessed measurement invariance across gender, which is a basic requirement for comparing structural paths between groups. If no invariance test was conducted, this omission should be acknowledged and discussed.

4b. Moreover, the figures illustrating the model and results are of poor quality, often stretched or pixelated, and not well integrated into the text. Figure captions are minimal, and figure references in the main text are sometimes out of order or missing entirely. The figures should be revised for visual clarity, and all statistical results (fit indices, parameter estimates, confidence intervals) should be fully and consistently reported.

  1. Discussion and Interpretation of Results

5a. The discussion addresses the main findings with some degree of psychological interpretation. However, it tends to overstate the causal implications of the results, despite the cross-sectional nature of the design. Claims that difficulties in emotion regulation “lead to” lower life satisfaction, or that digital game addiction “increases” emotion regulation problems, should be revised to reflect correlational rather than causal relationships.

5b. In interpreting the moderated mediation effect, the discussion remains somewhat superficial. The observed difference between males and females in mediation strength is consistent with existing literature, but the explanation remains generic and lacks theoretical grounding. Integrating frameworks from emotional development, socialization, or neurobiological models of gender differences would provide more interpretative depth. The suggestion that girls regulate emotions “more harmoniously” than boys, while plausible, would benefit from stronger empirical anchoring.

  1. Limitations and Future Directions

The limitations section includes important caveats, particularly regarding the cross-sectional design and reliance on self-report. However, it omits several critical issues. Most notably, the absence of measurement invariance testing undermines the validity of gender-based comparisons. I strongly suggest the authors to perform that (or similar) analyses. Furthermore, the restricted cultural context (Turkish adolescents from two schools) severely limits generalizability. Finally, no reflection is provided on the use of single-informant data, which may be particularly problematic in emotion regulation research. sFuture research directions should not only call for longitudinal or cross-cultural replications, but also suggest more refined modeling approaches (e.g., latent profile analysis, person-centered mediation), and mixed-methods integration to explore subjective experiences of gaming and emotion.

Author Response

Please see the attached file. All our responses to the reviewer 3 comments have been included in the PDF document.

Round 2

Reviewer 1 Report

Comments and Suggestions for Authors

Thank you to the authors for the changes made to their manuscript. All have improved it. A few changes remain.

  1. Although the authors introduce that gaming addiction is not an accepted psychiatric diagnosis, their work presents gaming addiction as one. It is insufficient to state in the Limitations that the authors realize it is not an accepted diagnosis, but they assume it is. Since it is not, the authors must refer to “excessive digital gaming” rather than “digital gaming addiction”, or call it “assumed digital gaming addiction”.

  1. It is especially problematic that the authors do not define “addiction”. This fundamental concept requires definition in the Introduction, supported by the most recent research.

  1. Citation 3 is the 2013 version of the DSM. There is a more recent publication. The most recent version of the DSM is the DSM-5-TR (Text Revision), published in 2022. Please cite this more recent version.

  1. There are a few citations that have no support of research published since 2021. Please provided supporting citations published since 2021 for these. They are listed here by line number:

    1. 107 (Baysak et al., 2020).
    2. 172 (Joormann & Stanton, 2016).
    3. 191 Kökönyei et al. (2019).
    4. 199 Cudo et al. (2020).
    5. 203–231 (Cudo et al., 2020; Bonnaire & Baptista, 2019).
    6. 241 (Phan et al., 2020).
    7. 318–319 (Anlı & TaÅŸ, 2018).
    8. 327 (Dağlı & Baysal, 2016).
    9. 369 (Tabachnick & Fidell, 2019).
    10. 452 (TaÅŸ, 2017).
    11. 457 (Loton et al., 2016).
    12. 490 (Zohreh & Ghazal, 2018).
    13. 533–534 (McRae et al., 2008; Whittle et al., 2011).

  1. There are several claims in the changes to section 2.4. Present study. However, there are only two citations, and the first is outdated. For each sentence that is a claim, there must be a citation to research published since 2021.

Author Response

Please see the attached file. All our responses to the reviewer 1 comments have been included in the PDF document.
Changes made in response to the first round of reviews are highlighted in red, and changes made in response to the second round are in purple (standard font). All modifications are tracked within the manuscript for clarity.

Reviewer 3 Report

Comments and Suggestions for Authors

I thank the authors and praise them for their revisions. My points have been appropriately addressed with honesty and transparency.

I would personally change the title since now it is quite complex and risks not to convey the essence of the manuscript. However, I leave such option to the authors/editors. 

Author Response

(The authors gave the same response as above.)

Round 3

Reviewer 1 Report

Comments and Suggestions for Authors

Thank you to the authors for the constructive responses to the suggested edits from the previous review. The result is a significant improvement to the manuscript. A few other problems are now recognizable.

Line-by-line suggested edits.
229 Although the authors avoid referring to problematic digital gaming as an addiction in most instances, it is called a “behavioral addiction” in this line. Change “gaming, a behavioral addiction observed” to “gaming is observed”.
242 Please cite research published since 2021 to support this claim.
246–247 Please cite these previous analyses.
254–255 Please cite these studies that treat gender as a secondary control variable.
256–257 Please cite the studies where a detailed analysis is lacking regarding the psychological impacts.
271 Please cite the source of Figure 1.
307 Please cite a study using the Game Addiction Scale for Adolescents-Short Form published since 2021.
315 Please cite a study using the Life Satisfaction Scale published since 2021.
360 Please cite the source of Figure 2.
378 Please cite the source of Figure 3.
386 Please cite the source of Figure 4.
418 Please cite the source of Figure 5.
439 Change “addicted to digital games” to “with problematic digital gaming behavior”.
440 Please cite a post-2021 study demonstrating that they spend most of their time playing games (and change “games” to “digital games”.
445 Change “digital game addicted” to “problematic digital gaming in”.
497–499 Please cite research published since 2021 to support this claim.
506 Change “eliminated” to “superseded”.
509 Please find a supporting reference for (Zysberg & Raz, 2019) published since 2021.
519 Please cite this neurobiological research.
530 Change “gamin” to “gaming”.
549 Please cite research published since 2021 for each of the three types of advanced analytic approaches.
552–555 Please cite research published since 2021 to support this claim.
555 Please post-2021 research on respondent bias.
563–564 Please cite the research that there is no global consensus on the clinical validity of problematic digital gaming.

Author Response

Please see the attached file. All our responses to reviewer comment 1 are included in the PDF.
Changes made in response to first-round reviews are shown in red, changes made in response to second-round reviews are shown in purple, and third-round revisions are shown in green. All changes are tracked within the manuscript for clarity.
